# The global costs of extreme weather that are attributable to climate change

Rebecca Newman[1] & Ilan Noy [2] ✉

Extreme weather events lead to significant adverse societal costs. Extreme Event Attribution (EEA), a methodology that examines how anthropogenic greenhouse gas emissions had changed the occurrence of specific extreme weather events, allows us to quantify the climate change-induced component of these costs. We collect data from all available EEA studies, combine these with data on the socio-economic costs of these events and extrapolate for missing data to arrive at an estimate of the global costs of extreme weather attributable to climate change in the last twenty years. We find that US$ 143 billion per year of the costs of extreme events is attributable to climatic change. The majority (63%), of this is due to human loss of life. Our results suggest that the frequently cited estimates of the economic costs of climate change arrived at by using Integrated Assessment Models may be substantially underestimated.

Extreme weather events have significant adverse costs for individuals, firms, communities, and regional economies. Based on the available data from the International Disaster Database (EM-DAT), the World Meteorological Organization[1] reports that there has been a sevenfold increase in the reported disaster losses from extreme weather since the 1970s.

While a part of this increase is due to increased reporting of disaster damage (especially in lower-income countries/regions or countries/regions that were previously more isolated), and because of increased exposure brought about by population growth and internal migrations to more exposed urban and coastal areas, a part of it is attributable to climate change. The most recent Intergovernmental Panel on Climate Change Report[2] notes it is virtually certain that there is a climate change component in the increase in reported disaster damage (at least of some types, with weaker evidence for others). The detection of anthropogenic changes in the frequency, severity, spatial location, and extent of extreme weather events is consequently important.

Extreme Event Attribution (EEA) is a methodological approach that examines the degree to which anthropogenic greenhouse gas emissions had changed the occurrence of specific extreme weather events that have indeed occurred. Using climate modeling tools, EEA quantifies the causal link between anthropogenic climate change and the probability and/or the intensity of specific extreme weather events by focusing on their specific circumstances and characteristics. EEA was first conceptualized by Allen[3], who, together with some co-authors, developed a method to analyze the contribution of climate change to the risk of an individual weather event that could be clearly defined and quantified. This approach was first implemented for the 2003 continental European heatwave[4]—an event that led to high mortality, especially in France.

The EEA methodology compares the probability of an event that occurred with the probability or intensity of the same event occurring in a counterfactual world without anthropogenic emissions. From a probabilistic perspective, a Fraction of Attributable Risk (FAR) metric is calculated to describe what portion of the risk of an extreme weather event occurring is the result of climate change. Methodologically, these probabilistic methods have been approached from both a frequentist or a Bayesian perspective[5], with possibly important consequences for the results thus obtained. We do not distinguish between these in our work here, given the relative paucity of Bayesian attribution work. The attribution approach based on FAR is known as the risk-based approach[6]. The alternative intensity approach calculates what share of a specific aspect of the risk (e.g., rainfall) was due to climate change. For instance, the 2017 Hurricane Harvey's climate change-induced economic costs were analyzed by both risk-based[7] and intensity-based[8] approaches.

[1]Reserve Bank of New Zealand, 2 The Terrace, Wellington 6140, New Zealand. [2]Victoria University of Wellington, 33 Bunny St., Wellington 6011, New Zealand. ✉e-mail: ilan.noy@vuw.ac.nz

The economic costs associated with extreme weather events can be measured in two ways: First, these include direct economic damage, which occurs during or immediately after the event. Using flooding as an example, where the hazard is heavy precipitation, direct economic damage may include destroyed housing and roads, or lost crops. However, an extreme weather event can also cause indirect economic losses. These are declines in economic value-added because of the direct economic damage. Examples of these indirect losses are wide-ranging. For the flood example, they could include microeconomic impacts such as revenue loss for businesses when access routes are inundated by floodwater, meso-economic impacts such as temporary unemployment in the affected area, or even wider-ranging macroscale supply-chain disruptions. These indirect economic losses can often spill out beyond the affected area, and indeed even beyond the affected country/region's borders. Indirect losses may also have long time lags, making them difficult to quantify. Generally, events that cause more damage will also lead to more losses, ceteris paribus. However, this relationship between direct damage and indirect loss is nonlinear, with high-damage events causing disproportionately many more losses as well. Because of these difficulties in quantifying indirect (flow) losses over a large variety of extreme weather phenomena in a large diversity of countries/regions and economies (thereafter referred to as countries for ease of exposition) and affected regions, this paper only focuses on the more easily quantified stock of direct damages.

By combining the data on direct economic damages, with the attributable share of the risk, we can quantify the climate change-attributable cost of these events. This attribution-based method for calculating the costs of climate change (from extreme weather events) differs fundamentally from other approaches to climate change cost estimation. Those other approaches use macroeconomic modeling embedded within climate models in various types of Integrated Assessment Models (IAM).

Given some of the data deficiencies in terms of temporal and spatial coverage, described in the following chapters, the purpose of this paper is not to produce a definitive quantification. At the current rate of progress in attribution research in meteorological science, we are still years away from obtaining a thorough and reliable global coverage of most socio-economically damaging extreme weather events. Our ability to measure the damage associated with these events is also far from being sufficiently comprehensive or accurate. Therefore, our aim is to demonstrate the use-value of the methodology, rather than reach an unimpeachable set of estimates. As better EEA studies and more thorough and exhaustive economic costs estimates for extreme events become available over time, and the method is refined, the precision of this approach's estimates will increase in tandem.

Here, we use the frequency approach to aggregate the global economic damage from extreme weather events attributable to anthropogenic climate change. For that, we collect data from all available attribution studies with a frequentist analysis and extract their FAR estimate. We then combine these FAR estimates with data on the socio-economic costs of these events. While our research is not directly comparable to the IAMs, it provides an additional form of evidence that suggests that most IAMs are substantially underestimating the current economic costs of climate change.

## Results

By examining the attribution information in conjunction with the cost information, we can calculate the climate change-attributed economic costs of extreme weather events. We first present these costs for the events in the master dataset, and then the results we obtain by extrapolating our findings to create a global estimate of these costs.

### Attributed costs for events in the dataset

From the 185 events in the dataset—a net of 60,951 deaths are attributable to climate change—75,139 deaths that occurred due to climate change in events that became more likely and 14,187 deaths in events that have become less likely due to climate change. The net statistical value of life cost attributed to climate change across the 185 events in the master database is United States (US) $431.8 billion.

Anthropogenic climate change is responsible for a net $260.8 billion of economic damages across the 185 matched events (without the extrapolation described in the next section). This is equivalent to 53% of the total damages recorded. More than 64% of the climate change-attributed damages are connected to storms, which is expected given the high damages from events such as Hurricane Harvey. Furthermore, 16% of the attributed damages resulted from heatwaves, while floods and droughts are each responsible for 10%, and wildfires account for 2% of the net attributed damages. Lastly, cold events, calculated as a fall in climate change-attributed damages, are responsible for only −2% of net attributed damages.

### Extrapolated global climate change-related economic costs of extreme weather

The results from extrapolating the attribution data across all global economic costs from extreme weather events are described below for the two extrapolation methods. Furthermore, we analyze the heterogeneity of globally attributed costs across time and event type.

The total climate change-attributed impacts, dictated by the respective extrapolation methods, have varying degrees of similarity. For heatwaves, the extrapolated estimates for deaths and damages are very closely aligned—less than one percentage point between the results from the two methods. For other event types, the disparities are wider. Notably, storm damages contribute substantially to attributed economic costs, making up over 60% of the total damages recorded in the EM-DAT extreme weather event dataset. There are two data comparisons where the estimates differ widely (greater than ten percentage points) between a global and continental approach: flood deaths (45%) and storm deaths (132%). These discrepancies in flood cost calculations occur because the FAR data points vary widely across attribution studies. These flood results are significantly impacted by a regional average FAR for floods in Africa of a decrease of 0.49, meaning that an estimated 49% of the decrease of risk of flooding in Africa can be attributed to anthropogenic climate change. Comparatively, the regional average FAR for floods in all other regions is positive, indicating an increase in risk resulting from climate change. This has a relatively large impact on the regional extrapolation results as floods cause a relatively high number of deaths in Africa and a comparatively low level of damages. This is a common pattern for disaster mortality and damage in low-income countries[9,10]. Moreover, the discrepancy between climate change-attributed deaths from storms is primarily driven by a regional average FAR in Asia (0.81) at least 20 percentage points higher than the FAR in all other regions. This has a notable impact on the results given the high number of storm-related deaths in Asia. However, it is important to recognize that the two noted regional average FARs that impact these results are calculated from a few data points—3 for floods in Africa and 1 for storms in Asia. Due to the lack of data relating to important event-type and continental combinations, the global average extrapolation method is used in all the tabulations described below, to minimize over-reliance on a small number of attribution studies.

The economic value of life lost to climate change-attributed extreme weather is obviously very dependent on the assumed value of statistical life. When using the US-UK mean Value of Statistical Life (VSL; as described in the data section), the climate change-attributed cost associated with mortality is a net US$ 1.79 trillion from the global extrapolation method.

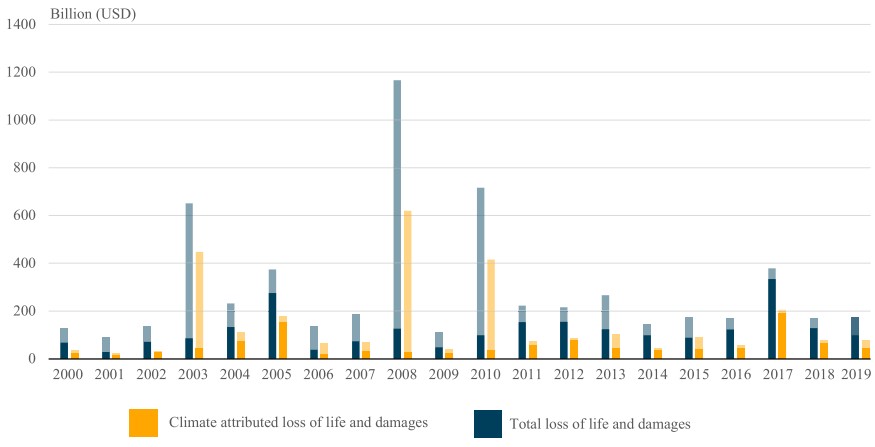

**Fig. 1 | Climate change-attributed loss of life and damages from extreme weather events.** These are the globally aggregated data for climate change-attributed impacts of disasters that were associated with extreme weather, using data collected from the Emergency Management Database - EM-DAT. Total costs represent the full estimate of the economic damages associated with an event, while the climate-attributed costs represent only the portion for which climate change is responsible. The combined bar represents the full cost, with the transparent portion representing the (statistical) lives lost and the solid portion are economic damages.

The estimated global cost of climate change over the 2000–2019 period is summed up in Fig. 1. These results are calculated using the global average FAR extrapolation method, which is less sensitive to singular studies than the regional average FAR approach. In aggregate, the climate change-attributed costs of extreme weather over 2000–2019 are estimated to be US$ 2.86 trillion, or an average of US$ 143 billion per year.

In an alternative calculation, in which we used the median FAR for each type of event, instead of the average (mean) FAR, the results are only larger: US$ 167 billion. This larger result is somewhat surprising, as our intuition was that larger events are more likely to be investigated in EEA projects, so that the median FAR should, in general, be smaller than the average FAR. This is not actually the case, suggesting this possible bias may be overstated.

This aggregate of US$ 143 billion, annually, is split across attributed human costs (statistical loss of life) of nearly $90 billion and economic damages of $53 billion per year. The distribution of costs is highly variable across years. The year with the lowest costs attributed to climate change is in 2001 at $23.9 billion, while the year with the highest climate-attributed costs is 2008 with $620 billion. The years in which costs reach high peaks—notably 2003, 2008, and 2010—are predominantly because of high mortality events. The events that drive these peaks are the 2003 heatwave across continental Europe; Tropical Cyclone Nargis in Myanmar in 2008; and the 2010 heatwave in Russia and drought in Somalia.

The aggregate result presented is subject to uncertainty given the limited number of data observations and the exploratory nature of this methodology. When considering a global FAR, for each type of weather event, one standard deviation below and above the mean the respective attributed cost per year is US$58 billion and US$228 billion, respectively. Storms drive the largest difference, given their contribution to absolute cost, however, the largest standard deviation is in flood events.

The peaks in climate change-attributed costs differ when we look solely at damages and exclude the statistical loss of life. The greatest peaks in monetary damages occur in 2017 and 2005. Storm events in the United States drive these—in 2005, Hurricanes Katrina, Rita, and Wilma together caused $123 billion in attributed damages, and in 2017, Hurricanes Harvey, Irma, and Maria were responsible for $139 billion in climate change-attributed damages.

Figure 2 shows how total and climate change-attributed costs are distributed across high (gross national income GNI per capita>USD 12,535), upper-middle (GNI per capita between USD 4046 and

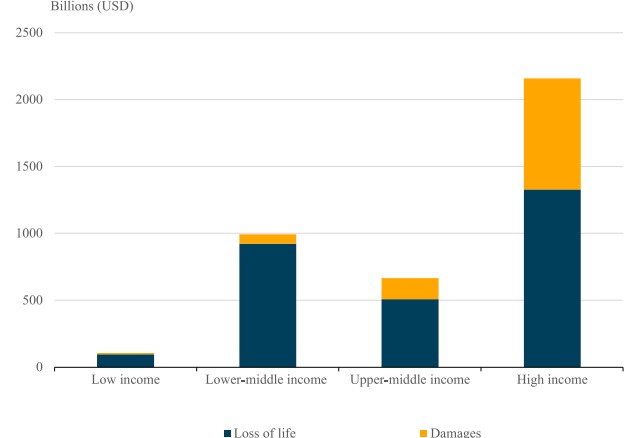

**Fig. 2 | Value for loss of life (VSL) and economic damage from extreme weather events by income group (2000–2019).** The aggregated mortality and economic damage costs for each country/region income group, using the 2020 World Bank's income classification.

US$12,535), lower-middle (GNI per capita between US$ 1036 and US$ 4046), and low-income (GNI per capita <US$ 1036) countries. This provides context for how different countries, especially the vulnerable ones, are being impacted by climate change-induced extreme weather. As per the available data, high-income countries have the highest climate change-induced economic costs at around 47% of the total. A few elements drive this, the primary being the United States having high asset exposure to storms.

However, the distribution of economic costs from extreme weather events across low to high-income countries is also likely a product of data availability and measurement. High-income countries have more resources and expertise to gather economic data when an extreme weather event occurs, while lower-income countries do not have this same level of resource availability.

These extrapolated estimates for the climate change-induced cost of extreme weather can be calculated as a proportion of gross domestic product (GDP), as shown in Fig. 3. Using the global average extrapolation method, the total economic cost, inclusive of damages and statistical loss of life, can be presented as a proportion of annual global GDP. This is not a direct comparison because GDP is a measure of economic flow, i.e., measured over a defined period, whilst damages

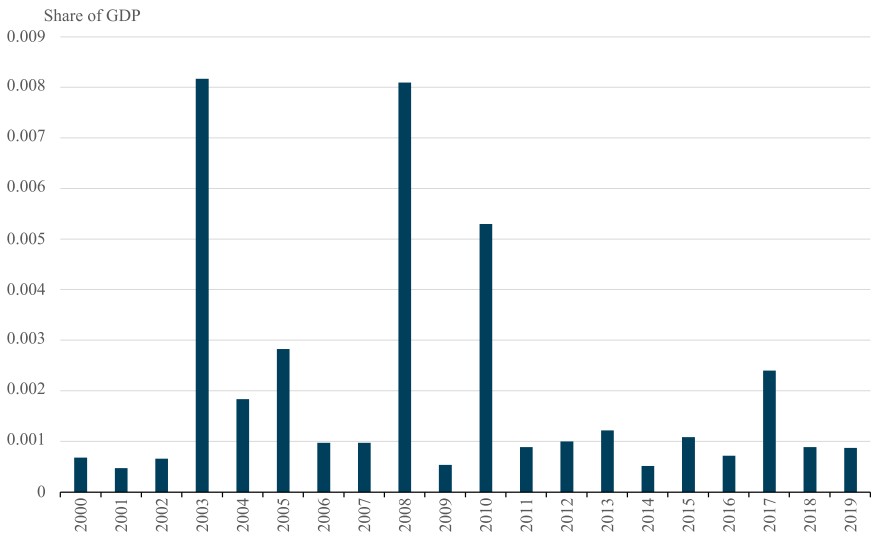

**Fig. 3 | Climate change-attributed loss of life and damages from extreme weather events as share of aggregate global Gross Domestic Product (GDP).** These are the globally aggregated data for climate change-attributed impacts of disasters that were associated with extreme weather, using data collected from the Emergency Management Database - EM–DAT. These represent only the portion which are attributable to climate change.

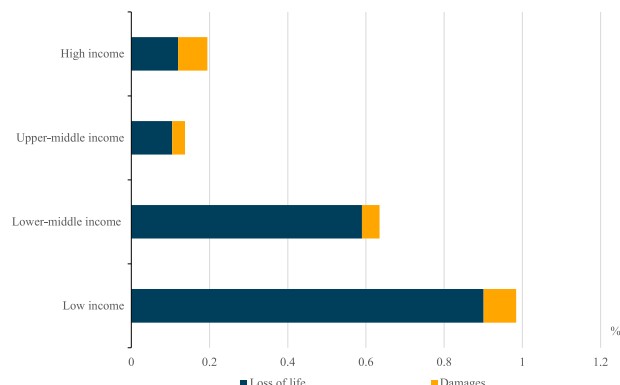

**Fig. 4 | Climate change-attributed costs of extreme weather events as a proportion of 2019 Gross Domestic Product by income classification.** This figures uses the 2020 World Bank's income classifications and are based on the average cost per annum over the 2000–2019 sample period.

and loss of life are a stock variable, i.e., measured at one point in time. It is, however, still a measure of the relative importance of these shocks on the affected economies. Climate change-attributed economic costs from extreme weather events vary between 0.05% to 0.82% of global GDP annually over the study period.

The limitations of comparing stock and flow withstanding, we can compare the annual average attributed costs for levels of GDP across countries at varying levels of development. This shows that low-income nations, as a cohort, experience the relative economic costs of climate-attributed extreme events to a greater degree—at near an average of 1% of GDP per annum, compared to 0.2% for high-income countries, as shown in Fig. 4. This differential is almost entirely driven by high levels of loss of life in lower-income countries, which may be the result of fewer early warning systems and safety procedures in place in these areas. In this context, we note our decision to use a uniform Value of Statistical Life across countries. That, of course, means that in lower-income countries, where mortality is highest, the relative importance of the loss-of-life measure is higher (as the value of assets is lower). Moreover, a smaller difference in economic damages may be the result of offsetting factors of higher-value assets in high-income countries, although buildings and infrastructure are likely to

be more resilient to weather events. This finding uncovers disparity in the costs of climate change, and a potential for inequality to become further entrenched due to greater extreme events.

## Comparing the cost estimates with integrated assessment models

There are several different approaches used to estimate the economic impact of climate change, with the attribution-based method of this research presenting an alternative option. The attribution-based method is an event aggregation approach; it therefore differs significantly from the macroeconomic methodology used in IAMs. Commonly, IAMs characterize damages as a polynomial function of the deviation of average annual temperature from pre-industrial times, as done, for example, in the Dynamic Integrated Climate Economy (DICE) model[11,12]. DICE approximates the damages from climate change, as a proportion of the global economy, according to the damage function shown in Eq. (1).

$$D(T) = \varphi_1 T + \varphi_2 T^2 \qquad (1)$$

Where $T$ is the change in global mean surface temperature above the pre-industrial threshold, currently estimated to be around 1.2 °C in 2020[13]. To allow us to compare the results from attribution to those of DICE, we used the parameters from the DICE 2016R model: $\varphi_1 = 0$; $\varphi_2 = 0.00236$, and the same temperature deviation data. This approach from DICE is not unique for IAMs. The Policy Analysis of the Greenhouse Effect (PAGE) model, which was used in the Stern Report[14], also calculates economic and non-economic damages from climate change using a polynomial function. However, PAGE uses regional temperature deviations rather than the global one[15].

From this basic calculation, as per the DICE model, the assessed global damages from climate change over 2000–2019 is estimated to be US\$ 4.04 trillion. Based on an aggregated event attribution approach, the approximation in this research is \$2.86 trillion, meaning the DICE estimate is ~40% larger. The comparative calculations of climate change costs from DICE and the attribution-based approach, by year, are shown in Fig. 5. However, these two metrics are not attempting to measure the same quantity, with two key differences:

First, the IAMs produce a measure of decline in economic flow (proportional to global GDP) while attribution-based estimates

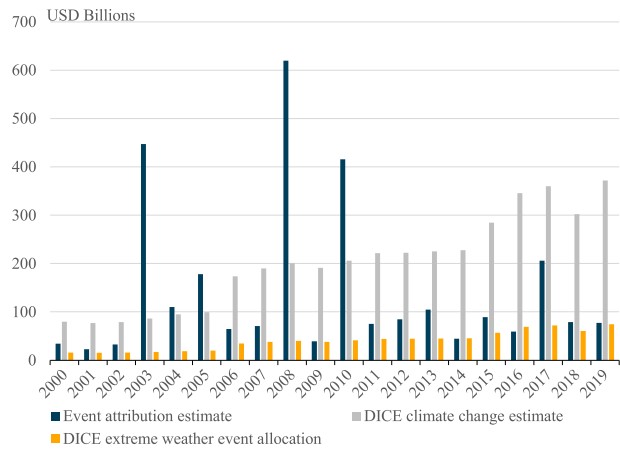

**Fig. 5 | Economic costs from climate change attribution and the Dynamic Integrated Climate Economy (DICE) model estimates.** The total cost calculated by the extreme event attribution method is shown relative to that calculated by the DICE damage function, as total in gray, and only the DICE estimates for extreme weather events (EWE) and other environmental damages in yellow.

measure loss in economic stock. This is the same distinction between damage and loss we described earlier.

Second, the attribution-based estimates solely measure the net economic cost of extreme weather events caused by anthropogenic activity, while IAM models attempt to estimate the overall annual loss caused by climate change. This should include extreme weather costs as well as many other types of costs and benefits from changing crop yields, ocean acidification effects, sea-level rise and its attendant impacts, environmental degradations and ecosystem disruptions, spending on adaptation, and many other types of impacts.

These factors limit the comparability of the IAMs measures and the attribution results. However, it is notable that extreme weather events are only one category of the damages that are, in theory, included in the DICE measure. The key limitation of IAMs, which is highlighted through comparison with the attribution-based approach, is that they account only for changes in average temperature rather than the change in temperature distribution, and specifically in the tail end of the distribution of weather-attributed. By focusing on the deviation in the average temperature, the IAMs fail to capture changes in extremes, plausibly the most important current impact of climate change.

Nordhaus acknowledges that DICE, and other IAMs, generally omit the impacts of extreme weather (as well as biodiversity, ocean acidification, catastrophic climate risks, and more). The solution he used to account for this limitation is to add 25% of the monetized damages in the DICE model[16]. This is a very subjective adjustment, which would assume that extreme weather accounted for a maximum of $0.8 trillion ($0.55 trillion would mean that extreme events account for the full value of the 25% adjustment to the DICE estimate) across 2000–2019, relative to the climate attribution-based figure of $2.86 trillion. This suggests a large underestimate that exhibits how DICE fails to accurately assess the economic impacts of climate change from extreme weather.

In addition, we can compare the attribution-based results to the Framework for Uncertainty, Negotiation, and Distribution (FUND) IAM, which is notably more complex than DICE. The FUND model differs from DICE as it calculates damages at a sectoral level, with nine sectoral damage functions operating across 16 regions of the world[17]. The key sector of interest in FUND, for this research, is the storm sector which is the only sector that is reflective of how climate change impacts the economic cost of extreme events. The FUND model calculates estimated damages (capital loss) and mortality for tropical and

extra-tropical storms. This is a more sophisticated inclusion of extreme weather event costs compared to the DICE approach. As an example, the total damages and mortality from tropical storms in FUND are calculated for each region using Eqs. (2) and (3).

$$Total\ damage = \alpha * GDP * \left(\frac{y_{today}}{y_{1990}}\right)^{\epsilon} [(1 + \delta * T)^{\gamma} - 1] \quad (2)$$

$$Total\ mortality = \beta * population * \left(\frac{y_{today}}{y_{1990}}\right)^{\eta} [(1 + \delta * T)^{\gamma} - 1] \quad (3)$$

In the FUND model, the key inputs in the damage function are temperature change over pre-industrial levels ($T$), per capita income ($y$), current damage as a fraction of GDP ($\alpha$), current mortality as a fraction of the population ($\beta$), and income elasticities of storm damage ($\epsilon, \eta$).

The MimiFUND web page, an accessible source for viewing the FUND model and results, estimates current damages from tropical cyclones as higher than the damages from extreme weather events calculated in the attributed results[18]. FUND calculates the current damage from tropical cyclones as, on average globally, 0.08% of GDP. Comparatively, the climate change-attributed damages from storms calculated in this research are 0.06% of GDP on average per annum. Further, climate change-attributed damages from all extreme weather events in the research equate to an average of 0.07% of GDP per annum. The difference in the FUND tropical cyclone estimation and the climate change-attributed costs of storms is an interesting comparison. It may be a discrepancy that can, to some degree, be explained by underestimated economic data recorded in EM-DAT the attribution estimates use. Furthermore, FUND estimates the current mortality from tropical cyclones to be on average 0.00015% of the population, while attribution-based results estimate that storms on average have a climate change-attributed mortality rate of 0.00009% per annum. These inconsistencies are illustrative of how, especially when data is lacking, it is beneficial to analyze multiple approaches to quantitative research—with the macroeconomic IAMs and event attribution techniques providing valuable contrasts.

## Limitations of the attribution-based approach

This research explores the potential of an attribution-based method for estimating the human-induced climate change costs of extreme weather globally. Although event attribution has been used to measure the climate change-related economic impact of individual extreme weather events before, this methodology has not yet been extended to a global approximation[7,8,19,20]. As such, this study does not provide a silver-bullet approximation of the cost of extreme weather events. There are important limitations of the attribution-based approach, primarily due to restrictions on the quantity and quality of data. These limitations are explored in detail below to highlight the progress required so as to improve these estimations.

When examining methodological limitations, we note that extreme event attribution is a young but rapidly expanding sub-field of climate science. The literature is limited, methodologies are continuously being refined, and the field's development faces some methodological and epistemological challenges. Notable limitations are the uneven geographical coverage of attribution studies and the lack of attribution studies conducted on several important classes of extreme weather events. These lacunae are significant, given the relatively small number of attribution studies conducted overall.

Extreme event attribution studies are more commonly conducted in high-income countries, with lower-income regions barely represented in the literature. In our database, only 8% of the attribution studies are conducted on extreme events in Africa, while over half of the events studied are in either North America (23%) or Europe (25%). In recent years, there has been a greater attempt to balance the

geographical distribution, particularly by the World Weather Attribution (WWA) network[21]. The WWA use the following human-based threshold to determine which events to consider for study: the event resulted in greater than 100 deaths, 100,000 people affected, or more than half of the total national population affected[22]. In contrast with an economic loss threshold, a human-based threshold leads to less bias against low-income countries where physical assets are of lesser value[23].

Still, extrapolation based on the total average FAR per event type leans over-proportionally on event probabilities from high-income regions (and China). The data gaps in Africa, South America, and Oceania, in particular, result in over-reliance on few data points in the calculation of a regional average FAR, or the use of an imperfect substitute (e.g., the global average FAR). This is a notable limitation because different regions of the world are subject to different climatic systems and environmental conditions. Consequently, the FAR for specific extreme weather events will differ by region and even more locally within countries. Improved geographical coverage of event attribution studies would improve the robustness of the methodology presented, especially if this allowed for greater granularity in the extrapolation method.

The second issue with event attribution data is the uneven spread of research across different event types. About a third of all attribution studies analyze the role of climate change in inducing heatwaves, the best-represented event category. Comparatively, storms, which are most important when considering the economic cost of extreme weather, make up only 8% of the studies in this dataset. One reason behind this discrepancy is the degree of difficulty associated with attributing different event types. Heatwaves, and similarly extreme cold events, generally result in the most reliable event attribution estimates as the direct thermodynamic effects for these events are comparatively straightforward[24]. In contrast, events such as agricultural droughts are caused by several compounding factors—such as precipitation, temperature, and soil moisture—making the attribution process significantly more complex. Cyclones are also complicated to model, which means that large-ensemble attribution studies of these storms have only become feasible in recent years, though a high computational cost for each simulation still persists[24]. As an example, Tropical Cyclone Idai that hit Malawi and Mozambique in 2019, and caused additional damage in Madagascar and Zimbabwe, was the costliest cyclone to have hit Africa with record-setting intense winds and rainfall, but even this event has not yet been analyzed in an attribution study.

Beyond the spatial and event-type coverage deficiencies, the framing of an event attribution study can induce large differences in how the role of anthropogenic emissions is quantified. Different framings would be appropriate for answering different questions[25]. One such example, which gained significant attention, was the 2010 Russian Heatwave. Two seemingly contradictory event attribution studies were conducted—one finding a negligible role of human-induced climate change, and the other identifying a fivefold increase in likelihood[26,27]. However, the framing of this event was central to this difference. The first paper analyzed the change in intensity, whilst the second analyzed the change in frequency. Moreover, subtle framing differences—such as whether attribution is conditioned on the background atmospheric conditions (e.g., El Niño-Southern Oscillation), or sea surface temperature conditions, or whether the counterfactual removes a single factor (greenhouse gas emissions) or all anthropogenic factors—can have a notable impact on the attribution quantification[6,28]. More reassuringly, recent examination of the variability of results due to different methodologies used in the EEA studies themselves suggest these results do not vary that much[29].

An attribution study must also define the spatial and temporal boundaries of the event being analyzed. These decisions ultimately impact the final FAR that is calculated[30,31]. As long as these definitions

of the event in the attribution study align well with the extent of the economic estimates produced by EM-DAT, this issue may not be as important. However, given the paucity of attribution studies, and the lack of detail about the geographical span of the EM-DAT data we could use, this was not always verifiably the case.

Commonly, event definition should reflect the main determinants of the event's impacts, as the authors seek to answer what role anthropogenic climate change played in creating the economic and societal impacts of an event[6]. For example, calculating a FAR using a single-day rainfall measurement (rather than, say a 7-day aggregate measure) may be preferable when a flood has caused devastation because of the short burst of intense rainfall that caused water to accumulate. For this study, attribution studies that define events based on the determinants of the most important human and economic impacts are beneficial. There of course can be multiple impacts, and these can be related to different event definitions. It is therefore not always clear which impact should be used when defining the event parameters. This is particularly salient for extreme events that are not meteorological in nature, such as flooding (hydrological) and wildfire (ecological), as these are also related to multiple climate parameters.

A closer geographical and temporal match between the FAR and economic impact data recorded in the dataset makes the calculation of attributed costs more reliable. However, events are not always defined in this way, as there may be barriers that prevent climate researchers from using such impact-based definitions. For example, it is often found that meteorological observational datasets are not extensive enough—across time or space—to allow an attribution study based at a specific locality or on a specific factor. Therefore, the event definition sometimes must deviate from the boundaries of the actual impacts to ensure the adequacy of data records[6].

Finally, it could be argued that attribution studies using the intensity approach, together with well-calibrated damage functions (that define the functional relationship between damage and the intensity of an event) might be a more appropriate input into our analysis. Two reasons led us to prefer relying on FAR quantifications. First, these are much more common in the attribution literature, allowing us to expand our sample of events. Second, we do not have well-calibrated damage functions, as these can be spatially and temporally specific, and are unique to each type of event (even different types of storms, for example, will necessitate different damage functions) and the social and built infrastructure exposures and vulnerabilities that differ substantially across locations[32].

One possibility to account for the uncertainty associated with the results is to generate a range of estimates based on the identified range of FARs. For example, we could make a similar calculation to the one we present, but based on the lowest (or highest) FAR identified for each category of hazard/region combination, rather than the average. While this will create a range of estimates, we are not able to conclude anything about the distribution function underlying this range. We therefore see such a range as potentially misleading, and prefer not to present these kinds of exploratory sensitivity analyses. However, since all of our data are posted publicly, an interested reader can explore this further, of course.

This research looked at events that became more or less likely to occur due to anthropogenic climate change. However, there may still be an embedded underrepresentation of events that have become less likely because of human-induced climate change; maybe because of publication bias, or because other factors that are associated with event selection. This is because attribution studies are typically conducted on major events, one that attracted the researchers' attention, and are not conducted at all on events that became mild because of climate change or have not occurred at all. Since there have been no recent occurrences for these events, it is impossible to quantify reliably their economic costs. There is no way to overcome this bias, but the available evidence seems to suggest that even before the main

impacts of climate change have started to be felt, the importance of these type of decreasing frequency or intensity events had been relatively less prominent than that of increasingly likely ones.

The economic data used to quantify the global cost of climate change-attributed extreme weather events in this study are subject to an additional set of limitations. They reflect the current best-available estimates, but there are possible limitations regarding the data's quality, coverage, and granularity.

The economic cost data used in this research underestimates the true costs of climate change over the study period. Most importantly, our estimates include only direct loss (damage) and not indirect loss ones. These later losses are difficult to measure. This is the case, among other examples, for productivity losses in a heatwave[33]. For example, the Australian Climate Council attempted a thorough approximation of the total economic impact of Australia's southwestern heatwave in 2009[34]. They estimated that the heatwave was responsible for up to AU$800 million in indirect financial losses— predominantly caused by power outages and transport system disruptions. This same event, as recorded in EM-DAT, detailed no asset damages at all. An inventory of events with the economic impacts differentiated into direct and indirect economic losses, at a bare minimum, would give decision-makers a better understanding of the wider economic impact of anthropogenic climate change[19].

The number of people affected by disaster events is recorded in EM-DAT. With the global average extrapolation approach, we found that climate change affected 1.4 billion people through extreme weather events between 2000 and 2019. Affected, in line with the EM-DAT definition, means requiring immediate assistance following the event. This could range from an acute need for life-saving medical attention and potentially sustaining life-long injuries, to the long-term provision of basic survival resources, or just supply of very short-term (hours or days) of emergency provisions. Clearly, there are significant economic costs associated with these affected people, including healthcare costs, costs of provision of other basic services such as emergency shelters, and potentially other longer-term welfare costs. However, given the extensive but imprecise range of costs that could be associated with someone being classed as affected, using a single monetary value for this group may be misleading. Therefore, these costs are not included in our calculations, but form an additional source of underestimation that is embedded in our results.

In addition, people can be adversely affected by an extreme weather event in ways that do not include requiring immediate medical assistance or basic survival needs. For example, people may suffer from mental health impacts (e.g., post-trauma), the loss of access to education, or the loss of their job if their place of employment is harmed. These will not be counted as having been affected, under the EM-DAT definition, yet suffer high economic loss. These costs are not captured in any available dataset.

While the limitations of this approach are significant, this research demonstrates how a more global approximation of the human-induced extreme weather event economic costs could be constructed. Each of the limiting factors described above has the potential to be reduced with more data collection and more research.

## Discussion

This research relies on two elements—the level of anthropogenic emissions and their consequential effect on climatic extremes (captured by the FAR), and the economic costs from extreme weather events. To minimize the climate change-attributed costs from extreme weather in the coming decades, there would need to be increased mitigation that will reduce the FARs, or an increased adaptation that will reduce the economic costs associated with extreme events, or preferably both.

Adaptation can make a considerable difference to the climate change-attributed economic impact of extreme weather events right now. Adaptation policies could include infrastructure development such as building flood protection or improving early warning signal systems for extreme weather events. A pertinent example of this, in our context, has been implemented in continental Europe, where the 2003 heatwave claimed upwards of 70,000 deaths, 55,400 of which were attributed to climate change. The extremely high mortality of this event shocked European countries into creating effective heatwave adaptation strategies to prevent a repeated high volume of deaths in the future. France, as an example, introduced a heat warning system that is triggered after three days of persistently high temperatures[35]. This system can enact the closing down of schools and public areas, the operation of a public heatwave helpline, and the opening of cool rooms in public buildings. This made a marked impact on the fatality of subsequent heatwaves. The heatwave in 2019 was hotter than that of 2003 in many locations, yet, in France, there were less than 1500 deaths, compared to over 19,000 in 2003. This clearly demonstrates how a well-designed and implemented adaptation policy can help reduce the climate change-attributed costs of extreme weather significantly. The results of this research, we hope, can provide an impetus to increase spending on climate change adaptation policies as it clarifies some of their benefits, in terms of avoided harm. It can also allow for better targeting of adaptation spending. This should ultimately help reduce climate change-attributed economic costs from extreme weather in the future.

For now, at the very least, more event attribution studies are needed, and the geographical and event-type representation of studies improved to align better with human impacts. This, in addition to better economic data, will allow the approximation of the global climate change-attributed economic cost of extreme weather to be improved, and thus form the basis for quantification of allocations through the Loss & Damage Fund. As such, this attribution-based method can also increasingly provide an alternative tool for decision-makers as they consider key adaptations to minimize the adverse impact of climate-related extreme weather events. This type of evidence can also fill, potentially, an evidentiary gap in climate change litigations that are attempting to force both governments and large emitting corporations to change their policies[36,37].

## Methods

Allen[3] suggested EEA as a method of comparing probabilities to quantify the contribution of climate change to the probability of an individual weather event occurrence. From this type of estimation, a FAR metric is calculated to describe what portion of the risk of an extreme weather event occurring is the result of climate change[6]. For this methodological approach, the weather is simulated under the current climate, and similarly, simulated under a counterfactual climate that is free from human greenhouse gas (GHG) emissions. This provides information on the degree to which climate change has altered the risk of event occurrence.

### Economic costs of extreme weather disasters

An extreme weather phenomenon by itself is not a disaster, but when a weather-driven hazard intersects with an exposed and vulnerable population (i.e., populations with characteristics that make them susceptible to adverse hazard impacts[2]), the extreme weather event becomes a disaster[2]. These events, when they occur, can cause a range of economic impacts. The Intergovernmental Expert Working Group on Indicators and Terminology Relating to Disaster Risk Reduction provides a set of relevant definitions. Firstly, a disaster can cause damages which occur during and immediately after the disaster. This is a stock amount that is measured in physical units and describes the total or partial destruction of physical assets, the disruption of basic services, and damages to sources of livelihood in the affected area. Relatedly, direct economic loss is the monetary value of these disaster damages, for example, the monetary value of totally or partially destroyed physical assets.

Secondly, disasters can cause indirect economic losses, defined as a decline in economic value-added because of direct economic loss (damages) and/or other disruptions caused by the disaster. These indirect losses can occur outside the disaster area and with a time lag and are measured as a flow variable (per unit of time). Indirect losses are more challenging to measure since they rely on developing a counterfactual (a without-a-disaster scenario). Finally, impact is the total effect of a disaster, including both negative effects (e.g., direct losses) and positive ones (e.g., indirect economic gains). Impact includes economic, human, and environmental impacts, including death, injuries, disease, and other adverse effects on human physical, mental, and social well-being. Some of these are intangibles that are rarely measured systematically after disaster events. This research will attempt to understand disaster impacts in aggregate and present them in terms of monetary valuation, referred to as the total economic cost. This is predominantly comprised of direct losses and the statistical value of life lost, given the limitations of the data collected in EM-DAT.

This approach, then, does not measure indirect losses. These may be significant. For example, the 2023 wildfires in Canada have imposed significant economic losses not only on Canadian cities impacted by the air pollution the fires generated, but they adversely impacted vast swathes of the densely populated North-East region of the United States (including New York City). None of the approaches discussed therein can account for these indirect losses, even though these could conceivably be orders of magnitude larger than the original damage wrought by these events (and were likely much larger in this specific case).

A challenge in aggregating damage data across international borders is the question whether damages have equal value in different countries. This problem is clearest for mortality. Typically, governments explicitly or implicitly attach a value of statistical life (VSL) to risk-of-mortality calculations, and these VSLs can be dramatically different across different countries. In low- and lower-middle-income countries (countries with GNI per capita <USD 1026 and USD 3995, respectively), a human life can be saved for a relatively much lower cost compared to upper-middle or high-income countries, so fiscally constrained governments in such countries typically use a much lower VSL in their policy decisions. However, we choose to use an identical value for a life anywhere. In this case, the flipside of this problem is to ask whether the monetary value of asset damage can be similarly aggregated internationally. Clearly, a $1 of value in a very wealthy country/region is a lot less consequential than a $1 in a very poor one. So as to clarify this further, we therefore present most of our results also separately for low, low-middle, high-middle, and high-income country groupings. Noy[9] and Wilson and Noy[38] provide more discussion of this issue and propose an alternative approach, which relies on a measure similar to Disability-Adjusted Life Year (DALY), instead of monetary values. However, the main argument for using a monetary unit of account herein is that our results are then comparable to others (for example, in the IAMs) and can also form a basis for Loss and Damage calculations.

Besides not measuring indirect losses, we also emphasize that the monetary measure we use (aggregating loss of life and direct damages) disregards any concerns about distributional consequences, even though these may very well have a significant impact on well-being. Our measure is purely utilitarian, but it could be enhanced with explicit assumptions about the specifications of the individual utility functions and the aggregate social welfare function. These kinds of approaches, however, will require a possibly controversial set of ethical and modeling choices.

## Using event attribution to estimate the economic costs of climate change

Allen[3] proposed that EEA enables differentiating economic losses from extreme weather between those that are caused by natural variability and those caused by past anthropogenic activity. Frame et al.[7]

suggested how this approach can attribute climate change-induced economic costs when both a fraction of attributable risk and economic cost inputs are available for a set of individual events. The approach they used is straightforward—multiply the fraction of attributable risk by the estimated economic costs. With some assumptions about aggregation and generalizability of the calculated FARs, this same process can be replicated across different types of economic impacts— including deaths, and even indirect losses—to provide individualized assessments of the climate change-attributed value of each of these impacts of extreme weather events. Frame et al.[20] estimated climate change-attributable insured costs of major flooding events in Aotearoa New Zealand based on the aggregation of attributed costs from 12 major flooding events. Some recent papers have looked at counting mortality and morbidity from heatwaves and attributing these to climate change[39–41].

Here, we aggregate all the relevant EEA studies (see details below), and their corresponding economic impact assessments and then extrapolate from these to obtain an overall estimate of the climate change-attributed impact of all recent extreme weather events globally, for which economic impact estimates are available. We then compare these estimates to some of the existing assessments of the current costs of climate change from the IAMs.

## Other methods for estimating the global economic impact of climate change

Most attempts to quantify the global impact of climate change use IAMs. Well-known, well-regarded, and equally well-criticized examples include DICE[12] and FUND[42]. The IAMs, typically, link the economic system with the climate system by using damage functions that express the economic impact of climate change as a function of a global or regional mean of annual mean temperature[43]. This, of course, captures the change in the mean, but not in the tail ends of the distributions of extreme weather[44]. Therefore, these models tend to include the costs of extreme weather using ad-hoc additional modification to the damage function, or they are omitted entirely[12,45–47].

Of course, comparing the IAMs to our approach using FARs is problematic, since the two are aiming to measure different quantities. IAMs model the economy and measure the decline in the flow of economic activity over time because of climate change—a very different approach to ours. Our argument is not a criticism of the IAM approach, per se, what we suggest is that the adjustments IAMs typically make to account for the impact of extreme weather events are significantly understated.

Given the limited availability of FAR studies, our approach cannot be applied across every extreme weather event. Consequently, the global application we pursue here relies on the extrapolation of known FAR values to other events for which there are no EEA studies, and a reliance on patchy economic data, to assess impacts (we discuss these data limitations further in the following sections). Van Oldenborgh et al.[22] argue that, with the current stock of EEA studies, we should consider the possible selection biases in the availability of EEA studies. Generally, events with higher human and economic impacts will be more likely to be analyzed, events in high-income regions and more densely populated areas are more likely to receive attention, and event types that become less likely because of climate change may be underrepresented in the analyses as well[22,48].

However, given the fundamental importance of empirical evidence to drive an informed climate change policy response, we use aggregation and extrapolation based on available knowledge, while acknowledging the limitations and inherent biases that might detract from the accuracy of such an exercise. Implicitly, we assume that the significant disaster events for which attribution studies are available are representative of the other damaging disasters of the same type, occurring in the same geographical region. Given the lack of a superior alternative, we see this is as an acceptable approach. We argue that all

current approaches to estimate the costs of climate change are limited by their methodological straightjackets.

Indeed, we argue below that the conventional IAM assessments is even less robust, and underestimates many of the most important impacts associated with extreme weather. This makes the exploration of an alternative and complementary cost estimation method fundamentally important, even if this method has its own flaws.

## Dataset collection and terminology

The fraction of attributable risk (FAR) is a metric that describes the portion of the risk of the extreme weather event for which anthropogenic climate change is responsible. When the risk of an event has increased due to anthropogenic GHG, it is calculated as shown in Eq. (4). This can be referred to as the fraction of attribution risk (FAR).

$$FAR = 1 - \frac{P_0}{P_1} \qquad (4)$$

$P_0$ = Probability of a climatic event without anthropogenic GHG present.

$P_1$ = Probability of the event occurring within the current climate system (with anthropogenic GHG).

A FAR value of 1 means that the event would not have been possible in the absence of anthropogenic climate change. While a FAR of 0 indicates that climate change had no influence on the probability of the event occurring[49]. More information on the data collection procedures we used is available in the Supplementary Information File.

To assess the economic cost of mortality, we utilize Value of Statistical Life (VSL) calculations; this is the standard approach in many policy decisions (for example, about road improvements for safety). The VSL describes a marginal rate of substitution between money and mortality risk in a defined period[50] and the VSL estimates differ very dramatically across countries[51]. The VSL we use here is an average of two VSL estimates used by the governments of the United States and the United Kingdom. The first is the United States Department of Transportation estimate for 2020, which sets the VSL at US\$ 11.6 million, which itself is an average of VSL estimates from across the academic literature[52]. The second estimate is from the UK Treasury, which assesses the VSL to be £2 million, estimated from average values from survey data looking at representative samples of the population[53]. According to Viscusi[54], the non-US median VSL is \$7.36 million (adjusted to 2020 USD). For this study, the benchmark result of US\$ 7.08 million per life lost is used, which incidentally is not very far from the non-US median reported by Viscusi[55]. For simplicity, and more importantly on equity grounds, we use this same VSL for deaths in every country/region, and every year, implying that death has an equivalent economic value regardless of the time and place in which it occurred.

Data for individual extreme weather events were matched, where both a FAR and economic data had been collected. These events were collated to form the dataset that provides the basis for our empirical analysis. The available data were refined to ensure the master dataset contained the best-available estimates for each included event.

When events with multiple attribution studies were available, the Scimago Journal Rank (SJR), in the year of publication, was used as a proxy for the research quality. The SJR impact factor was sourced from https://www.scimagojr.com/; it represents the rank of a journal's scientific influence and is calculated from a weighted measure of the citations a journal receives. The weighting is determined by the prestige of the publishing journal from which a citation originates[54]. We acknowledge, of course, that this procedure is not full proof, and papers that are sometimes considered better are published in lower-ranked journals. However, we wanted to use an algorithm that does not require any subjective judgment.

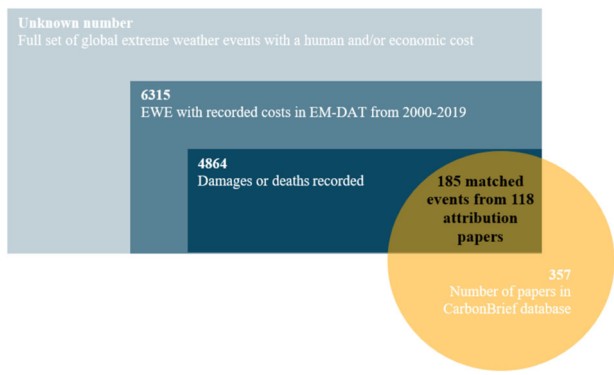

**Fig. 6 | The sampling method.** Of the whole 'universe' of extreme weather events (EWE), 6135 EWE are recorded by the Emergency Management Database (EM-DAT). If these 4864 recorded quantities for damages/deaths, but only 185 were matched with at least one of the 357 papers on attribution included in the CarbonBrief database.

A FAR measurement for a specific event is considered preferable if it comes from a higher SJR publication. For rapid studies conducted by the World Weather Attribution network, there was no recorded SJR as they are not refereed but are done by a large group of specialized climate scientists. Therefore, the average of the SJR impact factor scores for all other studies in the database was used as a rank for WWA studies when comparing them to others. When there are multiple attribution studies for the same event, with the same SJR, the preferred FAR was that with the closet spatial and temporal match to available economic data (as FARs can differ based on temporal and spatial event definition). When the scale is matched closely to economic data, the attribution of the cost will be more accurate.

The final dataset includes 185 events spanning 2000–2019. These events are gathered from 118 event attribution studies, as many attribution studies cover more than one event. Figure 6 depicts the hierarchical criteria applied in choosing the sample of attributed extreme weather events used to determine the FAR for analysis in this study, which is applied to the 2nd level of extreme events in the figure, that is 4864 events with human and/or economic costs recorded in EM-DAT.

## Methodological approach

Allen[3] states that "If [climate change] has trebled the risk over its 'pre-industrial' level, then there is a sense in which [climate change] is 'to blame' for two-thirds of the current risk…." (p. 891). This framing suggests that if anthropogenic climate change has made an extreme weather event three times more likely, then climate change is responsible for two-thirds of the economic cost caused by the set of similar events. Put differently, two out of each three events of the same class, and with the same calculated FAR, were caused by climate change, while the third would have happened even in a pre-industrial climate. Consequently, for each event (i) in the master database, we use Eq. (5) to estimate that individual event's climate change-attributed economic cost.

$$CCcost_i = FAR_i * cost_i \qquad (5)$$

Applying this approach to all events in the master dataset provides an estimation of the climate change-induced costs associated only with this specific list of events. To generate an estimate of the global cost of climate change from extreme weather events, we used the FARs from attribution studies in the dataset we collected and all the economic cost of extreme weather events across 2000–2019 recorded in EM-DAT. The events are limited to heatwaves, floods, droughts, wildfires, and storms, implicitly employing the EM-DAT definition of an extreme event. For inclusion in EM-DAT, an extreme event is one for which at

least one of the following three criteria must be fulfilled: (1) 10 or more deaths; (2) 100 or more people affected/injured/homeless; or (3) declaration by the country of a state of emergency and/or an appeal for international assistance.

Two extrapolation methods were used—a global average extrapolation method and a regional average method. The global average extrapolation method relied on obtaining an average FAR for each specific type of event occurring anywhere from the FAR results recorded in the dataset. This event-type average FAR was then multiplied by the economic costs and mortality of all the relevant events in EM-DAT over the 2000–2019 period. The average FARs are calculated from individual attribution studies in the dataset (118 observations) rather than the FARs from the 185 individual events. This is because some studies cover a large number of events. Calculating an average FAR with each event as an individual data point would lead to much greater weight being placed on a smaller number of multiple-event studies.

The regional average extrapolation method was conducted by calculating an average FAR per event type and per continent. This was, similarly, calculated from individual attribution studies rather than events. This regional average FAR was then multiplied by the relevant event-type and region-specific events in the EM-DAT database and subsequently aggregated. This (partial) accounting for differences in how climate systems influence extreme weather across different regions is clearly an advantage of the regional approach. However, there are no, or very few, FAR studies for some event-type and continental combinations. For example, only one study examined a heatwave in Africa, so a regional extrapolation result relies solely on this one study, creating potentially an over-reliance on one modeling approach.

Furthermore, where there are no available attribution studies, for example, on storms in Europe, the global average for that event type is used as a substitute to fill in this data gap. There are significant number of event-type-region combinations for which this global compromise was necessary. The difference between the two methods, therefore, is not as large as it probably should be. In the future, with a more extensive set of attribution study results, it would clearly be preferable to use an approach that distinguishes between types of events, their location (even within continental-size regions), and potentially even their magnitude.

**Extreme event attribution data**

Of the 185 extreme weather events, the risk of 154 of these events increased because of anthropogenic climate change, while another 24 events were associated with decreased risk, and the risk of the remaining 7 events was unchanged (FAR = 0). These events cover the period from 2000 to 2019. Notably, 77% of these events occurred after 2013, because EEA studies have been conducted increasingly frequently only in recent years. Given the rapid evolution of the EEA methodology, the dominance of more recent studies in our dataset means the FAR records used for the results reflect mostly the higher quality, recent EEA research practices. A significant number of events are recorded for 2015 because of the study by Zhang et al.[56], which covers a large spatial and temporal scale. With a larger dataset, we would have been able to place greater weight on more recent FARs, to reflect their higher reliability. However, our approach relies on a significant time span (in this case, 20 years) to account for the stochastic component in these event occurrences. If we were to place greater weight on, say, the last 5 years in the dataset, then our results will depend on what actually (stochastically) happened in those 5 years.

The geographical coverage of the matched attribution results included in the dataset is also important to note, as there are significant deficiencies in some regions. The studies cover Africa (10% of the studies), Asia (28%), the Americas (24%), Europe (20%), and Oceania (18%). North and South America are collected here as one grouping

because there are very few FARs calculated for South America. The events in South America make up only 5% of the total matched events in the dataset (7 events in the aggregate across all event types). The matched events span 52 different countries. Events in China, the United States, New Zealand, the Philippines, Japan, the United Kingdom, and Australia combined make up over half (54%) of the total dataset. Similarly to the time-series coverage, this is impacted by attribution studies covering multiple events in a defined region, including Zhang et al.[56] that covered 26 cyclones across the Western North Pacific.

By construction, the dataset contains six types of extreme weather events. Notably, 52% of the attribution results in the master database are associated with high-temperature phenomena—heatwaves, droughts, or wildfires (31%, 16% and 4%, respectively). The remainder are either hydrological events—specifically floods (37%) and storms (5%)—or cold waves (6%).

In the initial search, we identified 112 weather events with at least one associated FAR, but which did not have matching economic data. A majority of these (51%) were heatwaves, since the science of attribution is well-established for heat events but measuring the economic impact of heatwaves is challenging and is rarely undertaken. After all, the main impact of heatwaves, aside from mortality, is their indirect losses in the flow of economic activity which are substantially harder to identify and measure than damages (stocks). These under-measured heatwave losses include economic disruptions due to disturbed hydroelectricity distribution, transport failures, ongoing harm to agricultural crop yields and health, and harm to the natural environment[57,58]. Moreover, a further 25% of events without economic data are droughts—with the majority occurring in Africa—which is reflective of the geographically uneven distribution of disaster cost records between lower and higher-income regions.

All the events included in the dataset have at least one FAR associated with them. Of the 185 events, 47 have multiple relevant attribution studies. For each, the best FAR was selected based on the spatial and temporal match between the FAR study and the available economic data (see more details in the data collection section). The distribution of FAR attribution results is shown in Fig. 7. The peak at 0.3–0.4 is predominantly due to flood events—which make up 80% of the attribution results in this range. While 90% of the events with a FAR of between 0.7 and 1 are related to high temperatures, namely heatwaves, droughts, and wildfires. Interestingly, half of the attribution results with a zero or negative FAR are floods, while 32% are, less surprisingly, cold events. The remainder are three drought events with a zero FAR.

To allow a global extrapolation of climate change-attributed costs to be made, a global average FAR for each event type has been

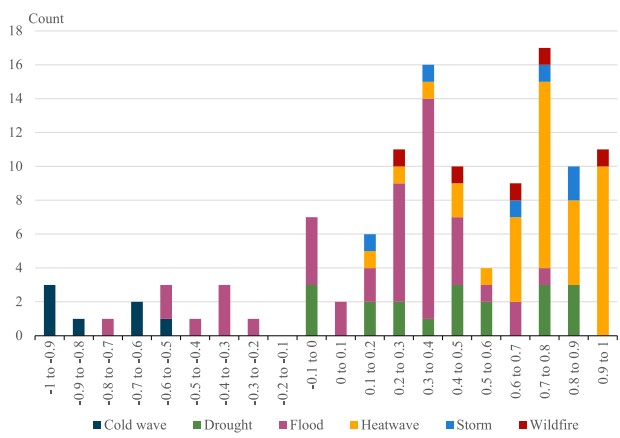

**Fig. 7 | Fraction of Attributable Risk (FAR) distribution across matched attribution results.** This figure shows the number of attribution events for each different FAR range, by the type of weather event.

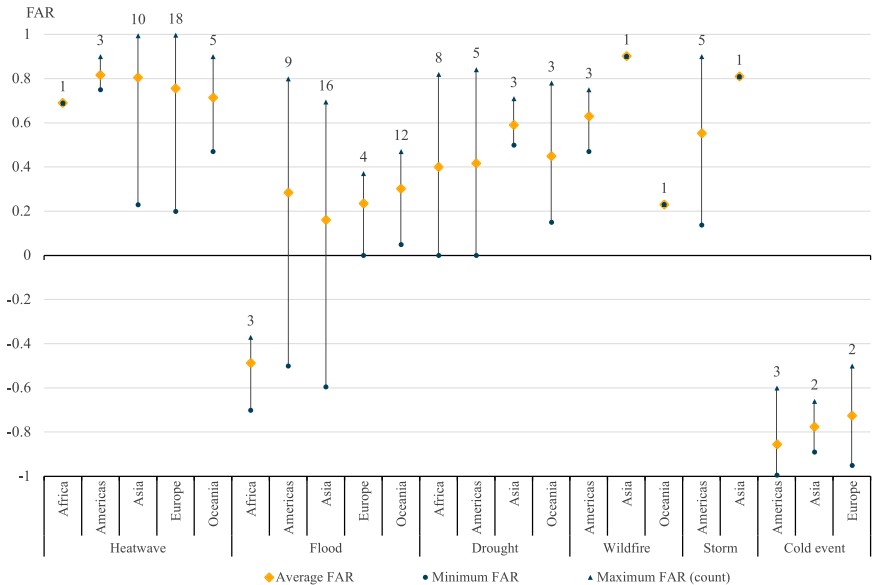

**Fig. 8 | Regional average Fraction of Attributable Risk (FAR) by event type.** This figure depicts the mean and the range of FARs matched, per weather type and by region. The number on the top of each vertical line represents the number of events for that region-type combination.

calculated. On average, 77% of the risk of heatwaves occurring over the study period is due to anthropogenic climate change. Floods show the greatest distribution range and are the only event type where attribution results span both increasing and decreasing risk due to climate change. The global average FAR for floods, however, is 19%. Similarly, as for droughts (44%), wildfires and storms each have a FAR of 60%; however, these are calculated from a small number of data points. Lastly, on average, cold events are calculated as having a decreasing risk (−79%) because of climate change.

An average FAR per-continent per-event type has also been calculated to reflect the lack of uniformity in the global climate system (Fig. 8). There were very few, or any, matched attribution results to form the basis of a regional average FARs in many continent-type combinations.

## Economic data

The following section describes the features of the economic cost data collected regarding the events in the master dataset that is based on EM-DAT—the most comprehensive global database of disaster impacts. EM-DAT defines a disaster event as one that surpasses a clearly defined threshold of damage caused (10 people dead, 100 people affected, an official emergency declaration, or a request for international assistance). As such, what is missing is a large number of small disasters, whose frequency may have risen because of anthropogenic climate change. An alternative database that does aim to capture these high-frequency low-impact events, Desinventar, is not available globally (Desinventar is collected and managed by the United Nations Disaster Risk Reduction Office).

While for disasters more broadly, EM-DAT records deaths, dislocations, people affected, and monetary damages, here we use only the mortality metrics and monetary damage (if these are recorded). The other components include such a diversity of outcomes (from dislocation lasting just a few days to permanent and significant physical and mental injuries), are often inconsistently collected, and are missing for many disaster events, that we have decided to ignore them in our analysis. As such, our aggregation only includes death and damage (as defined earlier).

In the dataset, 114 of the 185 events have mortality estimates. Thirty-nine of these events are responsible for at least 100 deaths each, nine events with more than 1000 lives lost, and four are responsible for

the deaths of over 10,000 people—a heatwave in Russia (>55,000 deaths), a drought in Somalia (20,000 deaths), a heatwave in France (>19,000 deaths), and a cold event in the United Kingdom (27,500 deaths). The total number of deaths recorded from the events in this dataset is 151,083, equivalent to a statistical value of life lost of US$ 1.07 trillion (using a VSL of $7.084 million).

Of the 185 events included in the dataset, 115 events have estimates for the economic damages caused. Across these 115 events, the total disaster damages stand at US$ 492.2 billion. The event with the highest damage recorded in this list of events is Hurricane Harvey in the United States in 2017, at US$ 100.3 billion. Harvey provides a useful example of some of the murkiness in definitions. EM-DAT classifies it as a tropical storm, and indeed the name refers to the hurricane. However, by far most of the damage was caused by the flood which the rainfall that came with the hurricane generated. The attribution papers that analyzed the event focused on the changing likelihood or intensity of the rainfall event, and not on the storm (measured and classified by windspeed). We follow the EM-DAT classifications, since these are available for all events, but note that these distinctions are not always immediately apparent.

Eighty-four of the events have estimated damages greater than US$ 100 million, and 8 of those are over US$ 10 billion. A small number of events in the dataset have insured loss estimates associated with them (48 out of 179). These data are heavily skewed to small number of countries, notably the United States, New Zealand, Australia, and Japan, as well as China. The restricted quality and quantity of data collection in low-income countries is one underlying reason for this, but it is also symptomatic of higher insurance penetration rates in high-income countries. Insurance costs from Hurricane Harvey in the United States and Hurricane Maria in Puerto Rico have the highest insurance payouts at US$ 31.7 billion each ($30 billion in 2017 US dollars).

For all of these events, indeed for all the events we analyze here, the causes of the damage are complex, and are not just due to the hazard itself. The 2010 heatwave in Russia is a good example. In Moscow, mortality mostly arose from air pollution from peat fires that occurred in the surrounding area. The fires occurred due to a combination of drought and heat, but also the legacy of a Soviet policy of draining bogs, widespread ignition by humans, and confusion and inaction following a new Russian policy shift that had just transferred

wildfire control from the national to the regional governments. Our analysis, however, assumes a *ceteris paribus* world in which all other pre-conditions still exist, but the amount of GHG in the atmosphere is pre-industrial. Clearly, one should view this counterfactual as a thought experiment, rather than a realistic scenario, since without the industrial revolution of the last 150 years, nothing in our world would have been the same.

## Reporting summary

Further information on research design is available in the Nature Portfolio Reporting Summary linked to this article.

## Data availability

All the data generated in this study have been deposited in the Harvard Dataverse database [https://dataverse.harvard.edu/dataset.xhtml?persistentId=doi:10.7910/DVN/N3ED1N]. The data are available under unrestricted access. More description of the data processing in this study are provided in the Supplementary Information File.

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

## Acknowledgements

We are grateful for very useful discussions with Dave Frame, Luke Harrington, Fraser Lott, Fredi Otto, Sue Rosier, Dáithí Stone, Belinda Storey, Eric Strobl, Michael Wehner, and many audiences in seminars and public talks. This research was supported by funding from the Whakahura Endeavour grant.

## Author contributions

I.N. conceptualized the paper. R.N. led the data collection and empirical analysis. R.N. and I.N. jointly wrote the paper.

## Competing interests

The authors declare no competing interests.
