## [Peer Review File · Nature Communications]

REVIEWER COMMENTS

Reviewer #1 (Remarks to the Author):

This manuscript titled “The Global Climate-Change-Attributed Costs of Extreme Weather” was designed to look at the impacts on climate change on economic losses from extreme weather events. From their results, they estimated that climate change caused around US\$ 143 billion per year of costs associated with extreme events during the last twenty years. This paper used a different approach at estimating these costs compared to previous estimates. According to this paper, the current methods greatly underestimate the costs associated with climate change. This manuscript provides important information the current economic costs associated with climate change. However, there are still some issues that need to be addressed before this paper is ready for publication.

First point, your cost estimates do not include total costs of health. This is an important thing to discuss, as these costs would likely change your estimates. Health studies have used attribution studies to estimate these impacts. I was surprised that this wasn't mentioned in their discussion.

Here are two studies that look at this:

Mitchell, D., Heaviside, C., Vardoulakis, S., Huntingford, C., Masato, G., Guillod, B.P., Frumhoff, P., Bowery, A., Wallom, D. and Allen, M., 2016. Attributing human mortality during extreme heat waves to anthropogenic climate change. *Environmental Research Letters*, 11(7), p.074006.

Puvvula, J., Abadi, A.M., Conlon, K.C., Rennie, J.J., Herring, S.C., Thie, L., Rudolph, M.J., Owen, R. and Bell, J.E., 2022. Estimating the Burden of Heat-related illness morbidity Attributable to Anthropogenic Climate Change in North Carolina. *GeoHealth*, p.e2022GH000636.

More thorough description of the methodology would improve the manuscript. I found it difficult to understand how the analysis was completed. It seems like the authors used the estimates of the contribution of anthropogenic climate change on extreme weather events to estimate the increased costs. From my understanding, a 25% increase in the event would increase the cost by a similar amount. Is there any evidence that is correct? It seems like it would be more appropriate to create a relationship between events and costs. From this, you could estimate the increase in cost with anthropogenic increase in severity.

It would be useful for the authors to edit the wording on the article. In a few places, the wording was more colloquial and less scientific. These issues could allow for misinterpretation. Line 77-80 and 175-177 are examples.

I recommend placing your data description before your methods.

More information is needed in the figure descriptions. I found it difficult to understand some of the figures and a more thorough description would have been helpful.

In addition, figures need improvement. I found some of the figures difficult to read. I highly recommend spending time improving the quality and readability of the figures. Most of the figures were difficult to interpret, but figure 8 was unreadable.

Line 306 what do you mean best journal? Is this an appropriate method? It would be nice to make this more repeatable. This seems very speculative.

Reviewer #2 (Remarks to the Author):

Thanks for the opportunity to review the paper “the global change-change-attributed costs of extreme events.”

This paper offers a major contributions to the knowledge on climate change impacts and natural disasters, and a very important one to inform very concrete discussions on policies and actions. The paper does a serious analysis.

While there are strong limit to the approach and methodology used, the authors are very transparent and provide the necessary caveats.

My assessment is that the paper can be published. I suggest below a few ideas to make the paper clearer and easier to read for the reader, plus one substantive suggestion.

The substantive suggestion is on the considered sample and how much it captures of the full risk. Maybe a summary table could be provided to help the reader understand how partial the analysis is. As I see it, you have (1) the full space of extreme weather events with damages; (2) a subset of those are recorded in EM-DAT; (3) a subset of those recorded in EM-DAT have data for human and economic losses. Then

there is 4th subset with events that have been subjected to an attribution study. I understand that the subset #4 has 185 event. The overlap between #3 and #4 is 115 event, but I'm not sure I understand how many event you have in subset #3, or the overlap between #2 and #3. Line 346 mentions \$492 billion for 19 years, which is clearly lower than the expected losses during the same period... In the paper, the subset #4 is extrapolated to the subset #3 to get to the \$143 billion number. My question is whether it's possible to extrapolate from subset #3 to the subset #2 or to the full set #1. And if it cannot be done (which is okay), I would suggest the authors could help the reader understand how partial the estimate (at \$143 billion) is. A table explaining these overlaps (maybe per hazards since it will be very different) would really help the reader interpret the \$143 billion number.

One side note on this: we know that EM-DAT is missing low-intensity events like the floods occurring annually in many cities in low- and middle-income countries; however these floods have huge impact on people and are driven by heavy precipitations that have increased with climate change. If the analysis misses these events, it's a useful caveat to provide.

I would recommend the paper systematically separate economic and human losses (including in the abstract when reporting on the total losses).

Another addition that the authors may want to consider is to use a range of FARs (e.g., when multiple estimate exist for a hazard, the min and the max, or the 25/75 percentiles?) to give a sense of uncertainty on the results. With the limits stated by the authors, it is particularly important to try to capture quantitatively the uncertainty in the final result.

More minor points

Lines 77-80 says that indirect losses are not included because it's too difficult. Line 128-130 says that the paper considers total economic costs. There is a contradiction here. Regarding indirect losses, I think it's probably useful to add that indirect losses tend to be large for large-scale disasters. Considering the importance of a few major events in the rest of the paper, that's probably worth flagging.

Line 175: this is an "acceptable" approach more than a "plausible" approach (what is a plausible approach?). More generally, the paper is a little defensive here, and I think it's not true to say that "all approaches are flawed". An approach is not flawed if it is useful even if it has limits (as is always the case). I would rather say that all approaches have their limits, and in a context of high uncertainty, it's important to use multiple and independent lines of evidence. The proposed approach here is a great complement to radically different approach used by IAM so it's extremely useful and a great contribution. But it's not a competition: even if IAM were great, it would be useful to explore other approaches.

I'm not clear on how the distinction between FAIR and FADR is useful for the rest of the paper. And I'm not clear why one would want to use a different index for decreasing risks. Equation (1) works for declining risks even if it's a little trickier to interpret. Or am I missing something? If risk is halved, then the FAIR is -100% (which means that the risk would have been 100% larger without climate change).

The question of how to make a "fair" metric is tricky here. Two points:

- Ethically, it makes a lot of sense to use the same VSL for everyone. But the right metric depends on how results are used. A lower VSL can be justified in low-income countries not because people are worth less (they are not!), but because in these countries one can save a life at a lower cost. For instance, small expenditures in health can save one statistical life in low-income countries while saving one statistical life in a high-income countries is much more expensive. If a VSL is used to decide if it's better to invest in disaster risk reduction or health systems, then using a lower VSL is the right choice (otherwise, one would invest too much in risk reduction and not enough in health systems). But the key is to separate the normative discussion from the practical (when the VSL is basically the Lagrangian in an optimization, not some estimate of how much people are worth).
- One could make the same argument for economic losses: while poor people are losing small amount of money, it affects their live very much. Losing \$1000 is a life-changing disaster for a poor farmer in a poor country.

I would expect extratropical and tropical storm to have very different results. Would it be possible to break down the storm category in two?

Could Figure 2 have a break down by hazard type so we see if the high FAR are for floods, heat, etc. ?

Line 360 says that climate change explains \$260b, which is 53% of total, but I guess this is not human + economic losses (since the total is \$492 + \$431 = \$923 billion). Can you clarify the total losses and the fraction, all together.

Figure 4: It took me a while to understand what the dotted line is. I would label it more clearly "Ratio of regional to global extrapolated FAR"

Line 404 this is where you could compare the total cost in your sample with various estimate of the expected annual loss, to give a sense of how partial the analysis is.

Line 441: Can you provide the % of GDP per country? Or at least separating LICs, LMICs, UMICs, and HICs?

Line 586: Interesting comparison of the results for intensity and frequency, but you may want to say a little more: how to explain the difference in findings? I can understand quantitative differences, but it's surprising to find nothing for intensity and 4 times for frequencies...

Line 618: I cannot agree more with the approach based on frequency. Using intensity and damage function would be impossible, with current damage functions!

Replies to reviewers are included in italics after each reviewer comment.

Reviewer #1:

This manuscript titled “The Global Climate-Change-Attributed Costs of Extreme Weather” was designed to look at the impacts on climate change on economic losses from extreme weather events. From their results, they estimated that climate change caused around US\$ 143 billion per year of costs associated with extreme events during the last twenty years. This paper used a different approach at estimating these costs compared to previous estimates. According to this paper, the current methods greatly underestimate the costs associated with climate change. This manuscript provides important information the current economic costs associated with climate change. However, there are still some issues that need to be addressed before this paper is ready for publication.

Many thanks for the kind assessment of our paper. Below, we detail our revisions based on your suggestions.

First point, your cost estimates do not include total costs of health. This is an important thing to discuss, as these costs would likely change your estimates. Health studies have used attribution studies to estimate these impacts. I was surprised that this wasn't mentioned in their discussion.

Here are two studies that look at this:

Mitchell, D., Heaviside, C., Vardoulakis, S., Huntingford, C., Masato, G., Guillod, B.P., Frumhoff, P., Bowery, A., Wallom, D. and Allen, M., 2016. Attributing human mortality during extreme heat waves to anthropogenic climate change. *Environmental Research Letters*, 11(7), p.074006.

Puvvula, J., Abadi, A.M., Conlon, K.C., Rennie, J.J., Herring, S.C., Thie, L., Rudolph, M.J., Owen, R. and Bell, J.E., 2022. Estimating the Burden of Heat-related illness morbidity Attributable to Anthropogenic Climate Change in North Carolina. *GeoHealth*, p.e2022GH000636.

Yes, we agree. We now include discussions of these papers. Unfortunately, we are only able to rely on the data available in EM-DAT. These data do not include much health data (other than mortality). Given our aim to cover all weather extremes, we have to rely on EM-DAT, as the only comprehensive global database that is available (unlike the papers cited above).

More thorough description of the methodology would improve the manuscript. I found it difficult to understand how the analysis was completed. It seems like the authors used the estimates of the contribution of anthropogenic climate change on extreme weather events to estimate the increased costs. From my understanding, a 25% increase in the event would increase the cost by a similar amount. Is there any evidence that is correct? It seems like it would be more appropriate to create a relationship between events and costs. From this, you could estimate the increase in cost with anthropogenic increase in severity.

We now improve the explanation of our approach. We rely on a change-in-frequency metric for climate change attribution; i.e., the identification of the change in the probability of an event happening because of climate change. As such, if 25% of the likelihood of an event is

because of climate change, we can assume that over a large number of realized events, 25% of them occurred 'because' of climate change (i.e., were fully attributable to climate change). This means we are not assuming that a 25% increase in the intensity of an event then leads to a 25% increase in its cost (this would be a more ambitious assumption, and one that would be more difficult to justify). We thus also do not need to develop a 'damage function' that links the intensity of the hazard, with the disaster's impacts.

It would be useful for the authors to edit the wording on the article. In a few places, the wording was more colloquial and less scientific. These issues could allow for misinterpretation. Line 77-80 and 175-177 are examples.

Done, we edited the whole article for clarity (we hope).

I recommend placing your data description before your methods.

Since a lot of our effort revolved around the data collection, we think the reader needs to know where we are heading with this data collection, before we 'bore them' with the technical details about this data. We therefore keep the order of sections, but edit the writing to clarify this sequence better.

More information is needed in the figure descriptions. I found it difficult to understand some of the figures and a more thorough description would have been helpful. In addition, figures need improvement. I found some of the figures difficult to read. I highly recommend spending time improving the quality and readability of the figures. Most of the figures were difficult to interpret, but figure 8 was unreadable.

We edited all the figures (indeed, almost all of them are now new).

Line 306 what do you mean best journal? Is this an appropriate method? It would be nice to make this more repeatable. This seems very speculative.

We now clarify this in footnote #10.

Reviewer #2:

This paper offers a major contributions to the knowledge on climate change impacts and natural disasters, and a very important one to inform very concrete discussions on policies and actions. The paper does a serious analysis. While there are strong limit to the approach and methodology used, the authors are very transparent and provide the necessary caveats. My assessment is that the paper can be published. I suggest below a few ideas to make the paper clearer and easier to read for the reader, plus one substantive suggestion.

Many thanks for the kind assessment of our paper. Below, we detail our revisions based on your suggestions.

The substantive suggestion is on the considered sample and how much it captures of the full risk. Maybe a summary table could be provided to help the reader understand how partial the analysis is. As I see it, you have (1) the full space of extreme weather events with damages; (2) a subset of those are recorded in EM-DAT; (3) a subset of those recorded in EM-DAT have data for human and economic losses. Then there is 4th subset with events that have been subjected to an attribution study. I understand that the subset #4 has 185 event. The overlap between #3 and #4 is 115 event, but I'm not sure I understand how many event you have in subset #3, or the overlap between #2 and #3. Line 346 mentions \$492 billion for 19 years, which is clearly lower than the expected losses during the same period... In the paper, the subset #4 is extrapolated to the subset #3 to get to the \$143 billion number. My question is whether it's possible to extrapolate from subset #3 to the subset #2 or to the full set #1. And if it cannot be done (which is okay), I would suggest the authors could help the reader understand how partial the estimate (at \$143 billion) is. A table explaining these overlaps (maybe per hazards since it will be very different) would really help the reader interpret the \$143 billion number.

This is a good suggestion. We added figure #1, to explain the partial cover offered by the EM-DAT data (and the attribution metrics). We now also discuss what extrapolation is possible (or plausible), and what is most likely not.

One side note on this: we know that EM-DAT is missing low-intensity events like the floods occurring annually in many cities in low- and middle-income countries; however these floods have huge impact on people and are driven by heavy precipitations that have increased with climate change. If the analysis misses these events, it's a useful caveat to provide.

Yes, low-intensity events are missed in the EM-DAT database (they are, in principle, included in Desinventar, but that database does not have global coverage). We now note this issue in footnote 17.

I would recommend the paper systematically separate economic and human losses (including in the abstract when reporting on the total losses).

Done.

Another addition that the authors may want to consider is to use a range of FARs (e.g., when multiple estimates exist for a hazard, the min and the max, or the 25/75 percentiles?) to give a sense of uncertainty on the results. With the limits stated by the authors, it is particularly important to try to capture quantitatively the uncertainty in the final result.

We are a bit reluctant to do this. If we include a range of estimates, it will be assumed that this range implies a statistical confidence level (say at the 95% confidence) that the 'real' estimate falls within this range. We don't really know that, and do not want to create a 'false impression' of statistical significance. We now explain this in the text at the end of section 8.1.

More minor points

Lines 77-80 says that indirect losses are not included because it's too difficult. Line 128-130 says that the paper considers total economic costs. There is a contradiction here. Regarding indirect losses, I think it's probably useful to add that indirect losses tend to be large for large-scale disasters. Considering the importance of a few major events in the rest of the paper, that's probably worth flagging.

Done. In section 2.1.

Line 175: this is an "acceptable" approach more than a "plausible" approach (what is a plausible approach?). More generally, the paper is a little defensive here, and I think it's not true to say that "all approaches are flawed". An approach is not flawed if it is useful even if it has limits (as is always the case). I would rather say that all approaches have their limits, and in a context of high uncertainty, it's important to use multiple and independent lines of evidence. The proposed approach here is a great complement to radically different approach used by IAM so it's extremely useful and a great contribution. But it's not a competition: even if IAM were great, it would be useful to explore other approaches.

We amended the discussion therein along the lines you suggest.

I'm not clear on how the distinction between FAIR and FADR is useful for the rest of the paper. And I'm not clear why one would want to use a different index for decreasing risks. Equation (1) works for declining risks even if it's a little trickier to interpret. Or am I missing something? If risk is halved, then the FAIR is -100% (which means that the risk would have been 100% larger without climate change).

We agree that this was rather confusing. We deleted this discussion.

The question of how to make a "fair" metric is tricky here. Two points:

- Ethically, it makes a lot of sense to use the same VSL for everyone. But the right metric depends on how results are used. A lower VSL can be justified in low-income countries not because people are worth less (they are not!), but because in these countries one can save a life at a lower cost. For instance, small expenditures in health can save one statistical life in low-income countries while saving one statistical life in a high-income countries is much

more expensive. If a VSL is used to decide if it's better to invest in disaster risk reduction or health systems, then using a lower VSL is the right choice (otherwise, one would invest too much in risk reduction and not enough in health systems). But the key is to separate the normative discussion from the practical (when the VSL is basically the Lagrangian in an optimization, not some estimate of how much people are worth).

- One could make the same argument for economic losses: while poor people are losing small amount of money, it affects their live very much. Losing \$1000 is a life-changing disaster for a poor farmer in a poor country.

Yes, we agree. We previously developed some of these ideas within the context of disaster damages in a set of papers on a Lifyears index (see Noy, 2016 and Wilson and Noy, 2023); and most recently used in the UNDRR's 2022 Global Assessment Report. We now briefly discuss these issues at the end of section 2.1.

I would expect extratropical and tropical storm to have very different results. Would it be possible to break down the storm category in two?

There aren't enough FAR observations for the these to allow us to separate these two categories of storms (in as much as they have different average FARs associated with them).

Could Figure 2 have a break down by hazard type so we see if the high FAR are for floods, heat, etc. ? Stacked colours

Done. In a new Figure 2.

Line 360 says that climate change explains \$260b, which is 53% of total, but I guess this is not human + economic losses (since the total is \$492 + \$431 = \$923 billion). Can you clarify the total losses and the fraction, all together.

The data was described confusingly previously. It is now clarified; both numbers refer to the attributable impacts. Thanks.

Figure 4: It took me a while to understand what the dotted line is. I would label it more clearly "Ratio of regional to global extrapolated FAR"

We decided that this figure is indeed confusing, and doesn't really contribute much, so we deleted it.

Line 404 this is where you could compare the total cost in your sample with various estimate of the expected annual loss, to give a sense of how partial the analysis is.

Done, we have re-done all the figures, so this part of the manuscript was completely re-written.

Line 441: Can you provide the % of GDP per country? Or at least separating LICs, LMICs, UMICs, and HICs?

Done, in new Figures 5 and 7. We think these comparisons are actually very important, so many thanks for these suggestions.

Line 586: Interesting comparison of the results for intensity and frequency, but you may want to say a little more: how to explain the difference in findings? I can understand quantitative differences, but it's surprising to find nothing for intensity and 4 times for frequencies...

We fully agree that this is interesting (and puzzling). These, however, were some of the earliest attribution studies produced, before a more precise algorithm was established; these estimates were therefore not done at the standard that is currently expected in this literature. Beyond that, as economists (i.e., users of attribution studies), we cannot really informatively comment on the specific methodological differences between these two calculations.

REVIEWER COMMENTS

Reviewer #1 (Remarks to the Author):

I appreciate the authors addressing the comments and changing the format of their article “The Global Cost of Extreme Weather that are Attributable to Climate Change”. Although the manuscript is much improved, some additional work would greatly improve the flow and quality of the paper. The authors should clearly discuss the limitations and assumptions of their approach. They should also add a section to their discussion on how their estimate might be still underestimating the total costs of climate. Here are a few more detailed suggestions.

The authors make the case that climate change attributed disaster damages are much larger than we would have otherwise guessed based on, among others, integrated assessment models (IAM). However, it should be noted that IAMs are modeling the entire economy and their numbers must not be taken out of context of the modeled relationship. Explaining this would be very important to improve the paper.

It is my opinion that the current study likely underestimates the true cost of climate change. Take for example a drought in the United States – the highest costs come from agricultural areas because of agricultural losses. However, other costs are not potentially included such as lives lost due to health issues that could occur because of the drought. For example, the drought may affect air quality that leads to increased deaths. You could also see overestimates of cost due to inflated property values in areas impacted by climate change. The manuscript would benefit with some discussion of these limitations.

Also, the authors make some assumptions that should be included in the discussion. For example, a study in 2000 is weighted as much as a study in 2019. If the authors add the caveat that the FRA methods are relatively younger, realistically a 2019 study should have a heavier weight than one in 2000. In this manuscript, there is not weight on the FRA so the authors assume the weight is constant across periods and disasters. While these may not need to be addressed, these assumptions should be mentioned.

In addition, there are still some grammatical errors and colloquial language in the manuscript. For example, you mention “rickety legs” in the results section. Removing these and a more thorough edit would be helpful.

The figures are improved, but it would be nice to define the acronyms in the figure. It would help with readability. Also, I think the description for Figure 4 was cut off or not added.

Reviewer #2 (Remarks to the Author):

I am happy with the changes made by the authors in response to my comments. Two last points:

While I understand their concern that a range of results may be misinterpreted as a probabilistic estimates, I still think that exploring how the final results vary when some of the key assumptions are changed in a reasonable range is informative and useful. Especially in this case, in which I would expect that the result is fairly robust to these assumptions.

The authors now explains properly their choice of a unique VSL. And they combine absolute economic losses with constant-VSL human losses, which mixes a pure monetary metric with a more normative choice. In the discussion on the direct and indirect economic losses, I would suggest to make it clearer that there are other metrics, and that real well-being losses are dependent on how the losses are being distributed within populations (so other metrics can include relative losses in % of income or welfare or wellbeing being defined with a utility or social welfare function).

I think the result from figure 7 (1% in LIC) is particularly striking, but probably due to the fact that with a unique VSL, human losses can very easily exceed GDP in low-income environment.

Many thanks for the suggestions. We followed through, and our detailed responses are below each comment, *in italics*.

Reviewer #1:

I appreciate the authors addressing the comments and changing the format of their article “The Global Cost of Extreme Weather that are Attributable to Climate Change”. Although the manuscript is much improved, some additional work would greatly improve the flow and quality of the paper. The authors should clearly discuss the limitations and assumptions of their approach. They should also add a section to their discussion on how their estimate might be still underestimating the total costs of climate. Here are a few more detailed suggestions.

The authors make the case that climate change attributed disaster damages are much larger than we would have otherwise guessed based on, among others, integrated assessment models (IAM). However, it should be noted that IAMs are modelling the entire economy and their numbers must not be taken out of context of the modelled relationship. Explaining this would be very important to improve the paper.

Yes, we add a paragraph (lines 187-192) that does exactly this, and explains a bit better the difference between our approach and the IAMs.

It is my opinion that the current study likely underestimates the true cost of climate change. Take for example a drought in the United States – the highest costs come from agricultural areas because of agricultural losses. However, other costs are not potentially included such as lives lost due to health issues that could occur because of the drought. For example, the drought may affect air quality that leads to increased deaths. You could also see overestimates of cost due to inflated property values in areas impacted by climate change. The manuscript would benefit with some discussion of these limitations.

Thanks for suggesting that. We now discuss this issue using a more contemporary and concrete example, the summer wildfires in Canada that have directed a lot of air pollution to the US North-East region (see lines 134-140).

Also, the authors make some assumptions that should be included in the discussion. For example, a study in 2000 is weighted as much as a study in 2019. If the authors add the caveat that the FRA methods are relatively younger, realistically a 2019 study should have a heavier weight than one in 2000. In this manuscript, there is not weight on the FRA so the authors assume the weight is constant across periods and disasters. While these may not need to be addressed, these assumptions should be mentioned.

We agree that it would have been better if we were able to rely more on more recent events. However, then our results would be more affected by the stochastic nature of what happened in these few years. We now discuss this issue in footnote 16.

In addition, there are still some grammatical errors and colloquial language in the manuscript. For example, you mention “rickety legs” in the results section. Removing these and a more thorough edit would be helpful.

We re-read the paper, and removed colloquial language where we found it (including those rickety legs).

The figures are improved, but it would be nice to define the acronyms in the figure. It would help with readability. Also, I think the description for Figure 4 was cut off or not added. *We removed the acronyms from the figures and made sure the description for Figure 4 is complete, as requested.*

Reviewer #2 (Remarks to the Author):

I am happy with the changes made by the authors in response to my comments. Two last points:

While I understand their concern that a range of results may be misinterpreted as a probabilistic estimates, I still think that exploring how the final results vary when some of the key assumptions are changed in a reasonable range is informative and useful. Especially in this case, in which I would expect that the result is fairly robust to these assumptions. *We agree that checking our assumptions is useful, though we still cannot really provide a statistical statement about the likely distribution of these estimates. We therefore re-did the calculations using the median FARs, rather than the means. This, of course, implies that now any very large or very small FARs no longer affect the estimates. Given what we previously discussed about the bias in conducting EEA studies on very obvious events (obviously related to climate change), we anticipated that these median-FAR calculations will lead somewhat smaller numbers. Somewhat surprisingly, the calculations based on median FARs are actually larger than those based on the mean FARs – \$167B instead of \$143B in average annual damage. We now discuss this in the text (lines 442-446).*

The authors now explains properly their choice of a unique VSL. And they combine absolute economic losses with constant-VSL human losses, which mixes a pure monetary metric with a more normative choice. In the discussion on the direct and indirect economic losses, I would suggest to make it clearer that there are other metrics, and that real well-being losses are dependent on how the losses are being distributed within populations (so other metrics can include relative losses in % of income or welfare or wellbeing being defined with a utility or social welfare function).

We agree, and we now discuss these issues (including explicit assumptions about individual utility functions and the social welfare function) on lines 153-159.

I think the result from figure 7 (1% in LIC) is particularly striking, but probably due to the fact that with a unique VSL, human losses can very easily exceed GDP in low-income environment.

Yes, we agree, we now note this in footnote 24.

REVIEWERS' COMMENTS

Reviewer #1 (Remarks to the Author):

I believe the authors did a satisfactory job in addressing the review comments and the article is acceptable for publication.